# Risk Factors, Diagnosis, and Management of *Clostridioides difficile* Infection in Patients with Inflammatory Bowel Disease

**DOI:** 10.3390/microorganisms10071315

**Published:** 2022-06-29

**Authors:** Livio Enrico Del Vecchio, Marcello Fiorani, Ege Tohumcu, Stefano Bibbò, Serena Porcari, Maria Cristina Mele, Marco Pizzoferrato, Antonio Gasbarrini, Giovanni Cammarota, Gianluca Ianiro

**Affiliations:** 1Gastroenterology Unit, Fondazione Policlinico Universitario Agostino Gemelli IRCCS, 00168 Rome, Italy; livioenricodelvecchio@gmail.com (L.E.D.V.); marcellofiorani94@gmail.com (M.F.); etohumcu09@gmail.com (E.T.); stefano.bibbo@policlinicogemelli.it (S.B.); porcariserena89@gmail.com (S.P.); marco.pizzoferrato1@policlinicogemelli.it (M.P.); antonio.gasbarrini@unicatt.it (A.G.); gianluca.ianiro@unicatt.it (G.I.); 2Department of Translational Medicine and Surgery, Università Cattolica del Sacro Cuore, 00168 Rome, Italy; mariacristina.mele@unicatt.it; 3Clinical Nutrition Unit, Fondazione Policlinico Universitario Agostino Gemelli IRCCS, 00168 Rome, Italy

**Keywords:** *Clostridioides difficile*, inflammatory bowel disease, Crohn’s disease, ulcerative colitis, fecal microbiota transplantation

## Abstract

*Clostridioides difficile* infection (CDI) and inflammatory bowel disease (IBD) are two pathologies that share a bidirectional causal nexus, as CDI is known to have an aggravating effect on IBD and IBD is a known risk factor for CDI. The colonic involvement in IBD not only renders the host more prone to an initial CDI development but also to further recurrences. Furthermore, IBD flares, which are predominantly set off by a CDI, not only create a need for therapy escalation but also prolong hospital stay. For these reasons, adequate and comprehensive management of CDI is of paramount importance in patients with IBD. Microbiological diagnosis, correct evaluation of clinical status, and consideration of different treatment options (from antibiotics and fecal microbiota transplantation to monoclonal antibodies) carry pivotal importance. Thus, the aim of this article is to review the risk factors, diagnosis, and management of CDI in patients with IBD.

## 1. Introduction

*Clostridioides difficile* (*C. difficile*) is a Gram-positive, anaerobic, spore-forming bacterium that represents an etiology of nosocomial diarrhea with increasing prevalence in the United States [1]. The transmission of *C. difficile* spores is established through the oral–fecal route, and they have a lifespan of up to a few months. Their recognized reservoirs are infected patients, asymptomatic carriers as well as other infected animals [2,3]. Asymptomatic colonization of *C. difficile* can be found in up to 15% of healthy individuals, yet only 25–30% of the colonized population develop an infection [3]. This being said, an increase in the incidence of *C. difficile* infection (CDI) in the general population has been observed in recent years [4].

The risk of progression from colonization to symptomatic CDI depends on the exposure to risk factors, such as contact with the healthcare environment, use of antibiotics, and advanced age [5], the latter also increasing the risk of disease severity and mortality. Compared to the general population, patients older than 65 years have a 5-to-10-fold increased risk of developing CDI [2,6]. Even though most cases of CDI can be found in hospitalized patients or in habitants of long-term care facilities, community-acquired infections are also increasing, reaching 35–48% of total CDI diagnoses [7,8].

Additional risk factors for CDI are gastrointestinal surgery, immunological incompetence caused by transplantation, malignant neoplasm, immunosuppressant use, chronic kidney disease, cardiac disease, and inflammatory bowel disease [2,9,10,11].

Patients with IBD can be analyzed as a sub-group and compared with the general population based on the age of presentation of the first CDI, infection recurrence, and several other risk factors. (Table 1). As evidence suggests, IBD patients tend to have their first episode of CDI at an earlier age through community-based transmission [12,13] and show a higher risk of having further episodes throughout life (13% vs. 7%) [12]. Patients with IBD are also associated with almost two-fold higher mortality (OR: 1.899 CI (1.269–2.840) compared to a patient with CDI in the general population [14] (Table 1).

Considering the differences mentioned above, the adequate management of CDI superinfection in patients with IBD becomes of uttermost importance. In this review article, our aim is to shed light on the latest evidence on this issue.

## 2. Risk Factors for Developing and for Severity of CDI in Patients with IBD

Several risk factors, one of the best-known being the overuse of antibiotics, contribute to the development of CDI in the general population either by directly killing or inhibiting the growth of the normal gut microbiota [20]. IBD is known to increase the risk of infection by nearly five-folds compared to the general population [12], and the likelihood of experiencing recurrences reaches almost 30% [12,21]. The medical therapy regimens of IBD, including 5-aminosalicylic acid formulations, corticosteroids, and infliximab as well as dysbiosis accompanying the mucosal inflammation, which is the hallmark of the pathology itself, are key factors that help explain the vulnerability in this patient population [15,21]. Additionally, the use of antibiotics, higher frequency of outpatient visits, which exposes patients to the microbial profile of the hospital, and a higher rate of comorbidities play a role in terms of CDI development [12].

On the other hand, the incidence of concomitant CDI accounts approximately for 5% of IBD relapses, as demonstrated in a study that explored the role of infections in triggering an IBD flare over 5 years [22,23], while based on a study with the recruitment of IBD patients in clinical remission, the prevalence of asymptomatic colonization by toxigenic *C. difficile* was nearly 8.2% compared to 1.0% in the general population [24].

The pathogenesis of inflammatory bowel disease includes many key points facilitating the colonization as well as the evolution of the process into a symptomatic infection by *C. difficile*, including abnormal intestinal immunity and alterations in gut microbiota, which in humans is one of the main protective mechanisms against the development of intestinal infections. Performing stool microbiology on specimens obtained from IBD patients in a flare-up, a study demonstrated that 10.5% of these relapses were associated with intestinal infections, half of them (and the most abundant one) being CDI [22]. In addition, recent evidence further supports the fact that CDI is the most common complication in IBD patients, leading to a flare-up [25,26,27].

Taking the general population into consideration, up to 30% of antibiotic-associated diarrhea and 50–70% of antibiotic-associated colitis are attributed to *C. difficile* infection, which instead is present in >90% of cases with antibiotic-associated pseudomembranous colitis [28,29]. As could be expected, in a recent meta-analysis [15], the use of antibiotics was shown to double the risk of CDI in the IBD population, while the risk regarding the role of antibiotics rises up to four-fold for the general population [17]. Taking into account that antibiotics are frequently used in IBD patients due to bacterial exacerbations or complications as a consequence of long-term immunosuppressive therapy [30,31], the correct evaluation of the clinical picture remains crucial.

As mentioned previously, the medical therapy of IBD plays a facilitating role in the etiopathogenesis of CDI in patients. Immune suppressants, such as biological therapies, double the risk of CDI among IBD patients (OR 1.65 [1.18, 2.30]) [15]. The higher rate of infection is not specific to *C.*
*difficile* but applies to other opportunistic infections as well [32,33]. Different degrees of risk can be attributed to various pharmacological agents used: vedolizumab, a gut-selective anti-inflammatory antibody, appears to increase the risk of CDI more than placebo [34], while there are discordant data on steroids, although in a 2018 cohort of 120 IBD patients steroid use was highlighted as an independent predisposing factor to CDI [15,35].

Colonic involvement, compared to a disease localization in the small bowel, is associated with an increased risk of CDI [15]. Disease severity and extension of the colonic mucosal damage remain to be key factors contributing to the infection to occur [36,37]. Indeed, CDI in Ulcerative colitis (UC) patients are 1.5 times more frequent than in Crohn’s disease (CD) [37] due to a higher proportion of colonic involvement in UC compared to CD [23], although in other studies, the incidence is the same [12,23]. The reason lying behind such a causality could be ascribed to a greater alteration seen in the gut microbiome and mucosal barrier permeability when the colon is affected by the pathology, but further insights and elaboration are needed to clarify this issue.

It is well known how principal human commensal bacterium, such as *Firmicutes* and *Bacteroidetes*, are involved in the formation of a protective gut barrier that prevents CDI and its recurrence [38,39,40]. Patients with IBD show a loss of diversity in their microbiota, with a reduction in *Akkermansia muciniphila*, a Gram-negative anaerobe with mucolytic properties [41], and also in *Faecalibacterium prausnitzii*, a short-chain fatty acid (SCFA)-producing bacteria with anti-inflammatory properties [42,43]. Moreover, alpha-defensins, which are known to have a neutralizing effect on the *C. difficile* toxin B, have an altered expression in this population, which exposes the population to an increased risk of CDI development [44,45,46]. These factors, together with a thinner mucus layer and an alterated distribution of mucins and phosphatidylcholine [47], could impact gut barrier integrity and consequently promote a bacterial translocation to deeper mucosal layers and thus, higher production of inflammatory cytokines.

Proton pump inhibitors (PPIs) in the general population are associated with a doubled risk of CDI, which can be explained by the greater rate of transformation of spores into vegetative cells, allowed by the reduced production of hydrochloric acid in the stomach [16,48,49]. The same data are not reproducible for IBD patients, where there is no such linearity. It has been hypothesized that gastric suppression does not amplify the susceptibility to CDI since IBD patients already present a remarkable dysbiosis in the gut [15,48,50,51,52,53].

## 3. Impact of CDI on IBD-Related Clinical Outcomes

CDI increases the risk of adverse outcomes in patients suffering from IBD. These patients have a higher risk of requiring a medical therapy escalation for disease control, longer hospital stays, increased risk of subsequent IBD flares, higher rates of surgery, and, finally, higher rates of IBD-related mortality in case of a concomitant CDI infection, latter risk being increased nearly four times compared to patients who have IBD alone [14,54,55,56,57].

The length of hospital stay in patients with IBD only is, on average, 3 days shorter when compared to IBD patients with CDI. Patients with concomitant CDI are less likely to respond to medical therapy for CDI and are more prone to have flares of underlying IBD, usually requiring a much more intensive therapy regimen (e.g., multiple drug treatment, biological agents). This population of patients also shows a higher likelihood of colectomy or other IBD-specific surgeries [14,15,37,54,55,56,57,58,59]. Although the risk of short-term colectomy does not appear significantly different between CDI with IBD and without IBD, the risk of long-term colectomy has been observed to be doubled for patients who have CDI and IBD [59,60]. These findings advocate for rapid and effective treatment of CDI in the IBD population.

## 4. Diagnosis

Clinical diagnosis of CDI in IBD patients could be challenging due to the similar profile of symptomatology shared by the two pathologies. The classical presentation of CDI includes abdominal pain with watery diarrhea, nausea, fever, and leukocytosis, often in people with nosocomial exposure or recent antibiotic therapy [61], even though such recent encounter or a CDI-justifying previous antibiotic use may be missing in patient history, which still would not allow excluding a CDI diagnosis [24]. It is well-known that the aforementioned symptoms are often present also in IBD flares. For this reason, when patients with IBD present with new diarrhea, CDI diagnosis should always be considered [37].

For an initial evaluation, a highly sensitive test is recommended, such as enzyme immunoassays (EIAs) for clostridial glutamate dehydrogenase, which is able to recognize all Clostridia strains, not just toxin-producing strains, or a PCR for toxin genes. To fully establish the diagnosis of colitis caused by *C. difficile,* a two-step approach is recommended [14,62]. If the highly sensitive test is positive, then a test with higher specificity, such as EIA for *C. difficile* toxins A and B, is the best choice. If the second test is also positive, then the diagnosis of CDI can be made, and in this case, treatment is mandatory.

In the case of a positive screening test but a negative second-level test, the patient is defined as colonized rather than infected, and an alternative diagnosis should be considered.

Endoscopic evaluation could be of use to recognize the typical mucosal involvement, namely pseudomembranous colitis, which is quite uncommon among patients with IBD [63] and to evaluate the extension and severity of inflammation. While in the general population, collection of biopsy samples is not mandatory, it can be useful in patients with IBD, considering they can be subjected to a higher risk of intestinal infections, to exclude other causes of diarrhea, such as cytomegalovirus infection, amebiasis, ischemic colitis, or IBD flares. Computed tomography of the abdomen and pelvis should be considered in patients with abdominal distention, fever, ileus, and/or hemodynamic instability to exclude bowel perforation or toxic megacolon, characterized by systemic toxicity and a colon dilation greater than 6 cm [64,65].

## 5. Treatment and Prophylaxis

Management of CDI is shaped by disease severity and frequency of recurrences. The infection is classified as non-severe, severe, or fulminant [1]. The presence of systemic signs, such as hypotension, shock, ileus, or megacolon, defines a fulminant CDI [1,66,67]. If factors correlated to a worse prognosis are present, including altered mental status, fever, cardiorespiratory failure, and lactic acidosis, an early surgical evaluation is indicated [65,67,68,69,70]. However, in patients who do not present systemic signs, severe and non-severe clinical pictures can be differentiated by laboratory exams: in particular, with leukocytosis (a white blood cell count of ≥15,000 cells/mL) or a serum creatinine level >1.5 mg/dL, the episode can be defined as severe [1].

Available treatment options for CDI can be classified into three groups: antibiotics, fecal microbiota transplantation, and monoclonal antibodies. (Table 2)

### 5.1. Antibiotics

Current medical therapy used in clinical practice consists of antibiotics, which are fidaxomicin, vancomycin, and metronidazole. (Table 3) Metronidazole inhibits nucleic acid synthesis, which eventually blocks protein synthesis and causes cell death [72]. Among antibiotics for CDI, metronidazole is the only one that, administered intravenously, is able to treat CDI. This is due to the pharmacological characteristics of metronidazole, which is excreted in bile and arrives at the colonic mucosa, as the oral form would. Metronidazole reaches higher concentrations in inflamed tissues; thus, once colonic inflammation decreases also antibiotic concentration declines. Since maintaining a constant mucosal concentration is not possible, its use is associated with higher CDI recurrence than other antibiotics, so it is no longer recommended as first-line therapy, except in areas where vancomycin and fidaxomicin are not readily available [27,73,74,75].

Vancomycin acts by inhibiting bacterial cell wall synthesis, having a bacteriostatic effect [76]. Taken orally, it has minimal systemic absorption, and its therapeutic effect can only be seen on enteric mucosa, reaching high concentrations in the colon [76]. Intravenous administration does not play any role in the treatment of CDI since it is heavily cleared by the renal route [76].

Fidaxomicin, instead, has a mechanism of action through the blockage of RNA synthesis. As vancomycin, it has minimal systemic absorption [77] and carries the advantage of having almost no effect on surrounding microbiota, in contrast to metronidazole and vancomycin. Since vancomycin also blocks sporulation, the lowest rates of recurrence are also observed when it is chosen for therapy [78,79]. Moreover, fidaxomicin was shown to be more effective than other antibiotics in terms of symptom disappearance [77,78,79,80,81] and was found to be capable of reducing environmental contamination as well [82].

The standard therapeutic approach to a first non-fulminant episode of CDI in IBD patients consists of oral vancomycin 125 mg four times daily for 10–14 days or, otherwise, fidaxomicin 200 mg twice daily for 10 days, which is significantly more expensive than vancomycin [83,84]. The last clinical practice guidelines of 2021, released by the Infectious Diseases Society of America (IDSA) (IDSA), suggest the use of fidaxomicin as the first-line treatment; however, they still consider vancomycin as a valid treatment alternative [71]. There are also interesting data collected from the use of long duration (21–42 days) oral vancomycin already from the first episode (tapered scheme), which appears to decrease the recurrence rates [85].

Fulminant CDI, besides requiring an early surgery evaluation, should be treated with oral vancomycin 500 mg four times daily, through a nasogastric tube if necessary, and intravenous metronidazole 500 mg three times. Vancomycin administered rectally should be considered as an add-on medical strategy.

Patients with IBD have an increased risk of CDI recurrence, up to 30% higher than the general population [21]. The first recurrence of CDI, identified by stool testing, should be treated with different antibiotic regimens, depending on the drugs used during the first episode. If oral vancomycin was used as starting treatment, for the second episode, a pulse-tapered regimen of oral vancomycin, including 125 mg daily for 14 days followed by 125 mg twice daily for 7 days, after this 125 mg once daily for 7 days, and then once every 2–3 days for 2–8 weeks, is indicated [1]. Another option for these patients is oral fidaxomicin 200 mg twice daily for 10 days [80]. Patients treated with oral fidaxomicin or metronidazole during the first episode can receive a standard 10-day regimen of oral vancomycin for the first recurrence. For the second or further recurrences of CDI, there are no clear recommendations for patients with IBD. Although oral vancomycin pulse-tapered regimen or oral fidaxomicin is not contraindicated, fecal microbiota transplantation (FMT) should be taken into account. On the other hand, there is no recommendation for the use of rifaximin in combination with vancomycin for the second or subsequent CDI recurrences in this population of patients [71].

### 5.2. Faecal Microbiota Transplantation (FMT)

An innovative therapeutic strategy for recurrent or refractory CDI is FMT. This approach appears to be a safer and even a better-tolerated treatment option for recurrent CDI in patients with IBD without causing a statistically significant increased risk for IBD flares [86,87,88].

FMT includes the process of transferring fecal bacteria and other microbes collected from a healthy individual into a recipient’s gastrointestinal tract. In a retrospective study, FMT had a curative effect on 74.4% of patients with IBD who suffered from recurrent CDI, compared to 92% of patients without IBD [87]. However, in another study, response to FMT for recurrent CDI was similar (nearly 90%) between IBD and non-IBD populations [89].

The different routes of delivery influence the efficacy rates of FMT [90]: for example, the duodenal and enema delivery have lower rates of complete clearance compared to the delivery via colonoscopy. However, FMT remains an equally effective, if not superior, therapeutic approach regardless of the route of administration when compared to antibiotic treatment [91,92]. Another route of delivery is the use of fecal capsules, and this approach was demonstrated to be not inferior to colonoscopy to cure recurrent CDI in the general population [93]. A lower dose (10 capsules in a single administration) seems to have the same effect as the higher dose (30 capsules) in terms of treatment outcome [94]. Lately, live microbiota preparations have been developed with promising results but must be further examined for the IBD population [55]. FMT through colonoscopy, compared to standard antibiotics therapies, may be associated with higher efficacy, but it also seems to be the most cost-effective for rCDI [95]. These facts could support the use of capsules in the near future, even though their use should be further elaborated in-depth for IBD patients.

In a recent cohort study from our center, conducted on 18 patients with IBD who received FMT due to rCDI, sequential FMT appeared to be a successful strategy in this sub-population and could be adopted as a priori protocol because of the higher risk of severe CDI [13,21] and recurrence that these patient bear. Worthy of note, in the same cohort, an amelioration of IBD disease activity itself after FMT was observed as well [96].

Another experimental protocol is ribotype-guided fecal microbiota transplantation, which consists of FMT in recipients colonized by particularly virulent strain, such as ribotype 002 strain. This method, used in a Chinese cohort, appears to save quality-adjusted life years (QALYs) compared to a standard course with vancomycin [97].

### 5.3. Monoclonal Antibodies

Another therapeutic strategy for the prevention of recurrent CDI among high-risk populations, such as people who suffer from IBD, is the use of Bezlotoxumab, a monoclonal antibody against the *C. difficile* toxin B, has been approved in 2016.

In a post hoc analysis of high-risk subpopulations, a single infusion of bezlotoxumab during the first antibiotic regimen for CDI resulted in a 27% reduction of recurrence within the next 12 weeks among patients with IBD [98]. Bezlotoxumab is recommended for any second recurrence within 6 months and also in primary CDI, in particular subgroups, e.g., in patients older than 65 years, with a severe clinical presentation, in infection caused by certain virulent strains (ribotypes 027/078/244), or in patients under immunosuppressive therapy, such as with infliximab [71].

Another potential utilization of bezlotoxumab is with FMT to boost FMT efficacy, especially in high-risk populations. In this regard, there is a phase 2 trial, called ICON-2, that compares FMT and Bezlotoxumab to FMT and Placeboin patients with IBD and CDI from four tertiary care FMT referrals centers [86]. Despite the hopeful results, the use of bezlotoxumab should be evaluated carefully in patients with congestive heart failure (CHF), as some reported cardiologic exacerbations after infusion [98].

In addition to bezlotoxumab, another human monoclonal antibody was designed for the prevention of rCDI, but according to a recent meta-analysis, it was not superior to placebo [99].

### 5.4. Probiotics, Prebiotics, and Future Strategies

Probiotics’ role in the prevention of CDI is controversial, and their routine use is not recommended by the European Society of Clinical Microbiology and Infectious Diseases (ESCMID) [62,66]. A Cochrane review has demonstrated that short-term use of probiotics could reduce CD-associated diarrhea (CDAD), mostly during antibiotic treatment and in patients who are not immunocompromised or severely debilitated [100]. Different probiotics may possess different mechanisms of action (Figure 1) that can be useful in overcoming CDI. In particular, *Saccharomyces boulardii* can secrete proteases capable of degrading CD toxins and reducing enteric CD-binding receptors [101,102]; *Lactobacillus rhamnosus* may have a role in stimulating intestinal immunity and in reducing intestinal permeability [103]; *Lactobacillus lactis*, on the other hand, has been described to possess the capacity to lyse CD [24]. In vitro studies have shown that various probiotics could have an effect on *C.difficile*, e.g., *Bacillus thuringiensis* produces thuricin CD, a bacteriocin that is able to form membrane pores against *C. difficile* strain, while *L. reuteri* is able to produce reuterin, a large spectrum antimicrobial peptide that has an inhibitory effect on *C. difficile* [104]. A non-traditional probiotic is a preparation containing non-toxigenic *Clostridium* species, such as *C. scindens* [105], which is able to compete with toxigenic *C. difficile*, demonstrating its effect by replacing that strain as well as playing a role in the conversion of primary bile acids into secondary ones, which inhibit *C. difficile* vegetative growth, since primary bile acids are crucial for spore development and evolution [105,106]. Data from the general population seem to confirm a reduction of recurrences with the use of probiotics [105], but this formulation has not been tested in IBD patients yet, and for its introduction to routine use in clinical practice, further evidence would be necessary. A randomized controlled study exists, which was conducted on 142 patients with CDI supplement of oligofructose (12 gr/day) and has also shown to significantly reduce the recurrence rates [107]; however, the population was not constituted of IBD patients.

From the point of view regarding prophylactic treatments, the development of a vaccine, just like its “relative” bacterium, *C. tetani* is a promising outlook for the future [108]. Most of the vaccines target CD toxins, but this field is still in its preliminary steps [106,108,109,110,111].

Another interesting therapeutic option for the future is the use of *C. difficile* phages. Phage therapy with CD140 has been demonstrated in animal models and in vitro models to reduce *C. difficile* virulence and to protect from a second re-infection. Another use of phages, called “phage cocktails therapy”, was able to weaken *C. difficile* strains resistant to a single phage [112]. Phages could be used in the future as prebiotics to improve the efficacy of existing therapy. Other researchers have focused on using just the tail-like particles, called bacteriocins, that could be more effective in targeting a variety of *C difficile* strains [112].

The study and manipulation of mucosal biofilm are becoming another field of interest. Because of their baseline dysbiosis, patients with IBD have a higher presence of polymicrobial invasive biofilms [113]. This assemblage of surface-associated microbial communities, composed of exo-polysaccharide (EPS) and various messengers used for their chemical communication called quorum sensing, causes resistance to various treatments, including antibiotics and the efficacy of host immune responses [114,115,116]. In other words, recurrence could be caused by resistance to antibiotic therapy or the presence of a biofilm [102]. It has been demonstrated that *C. difficile* is capable of producing biofilms [117,118,119], and it is hypothesized that it could protect itself from the immune response and other antimicrobials [120,121,122]. Vancomycin and metronidazole in sub-inhibitory concentrations can increase biofilm formation in some *C. difficile* strains, and these antibiotics are associated with increased resistance in the clinical isolates, as well as with a higher rate of clinical recurrence of CDI compared to other treatments [123,124]. In particular, certain strains of *C. difficile*, growing as a part of the biofilm, can survive up to 12-fold more than planktonic cells when exposed to vancomycin [123,124] and are 100-fold more resistant when exposed to metronidazole [120]. Fidaxomicin, on the other hand, is effective in killing bacteria and reducing bacterial spores within the biofilm [123] and coherently is also more successful than other antibiotics in reducing recurrent CDI incidence.

Another study tested various antimicrobials, such as thuricin CD, teicoplanin, rifampicin, tigecycline, and nitazoxanide, to act on biofilms formed by different *C. difficile* strains, showing that combined antimicrobial treatments were more effective than single drugs against some strains and that there are variations in the insensitivity of the biofilm produced by the different strains of *C. difficile* regarding several antimicrobial treatments [125]. This evidence suggests that the role of biofilms in *C. difficile* persistence or relapse after antibiotic treatment should not be neglected, and future investigations could lead to new effective treatments [126].

## 6. Managing IBD in Patients with IBD and CDI

There was a strong consensus among all the latest European and American guidelines, particularly focusing on the use of diagnostic triage every time there is a suspected flare of IBD. In this case, they recommended always performing a biochemical assessment on a stool sample and a high sensitivity test for *C. difficile* [31,127,128].

Especially in patients with acute severe ulcerative colitis (ASUC) (>6 bloody stools per day and at least one among these systemic toxicity signs: temperature > 37.8 °C, pulse > 90 bpm, hemoglobin < 105 g/L or C-reactive protein >30 mg/L) and pending results of stool exams, British Society of Gastroenterology recommend to not delay treatment with high-dose intravenous corticosteroids (methylprednisolone 60 mg daily or hydrocortisone 100 mg 6-hourly) and start therapy right away [31]. Once the CDI is confirmed, specific antibiotic therapy should be initiated, and corticosteroids should be suspended [31].

The clinical symptoms and signs must be evaluated for the next 3–4 days after the beginning of antibiotics, and if no improvement occurs, an endoscopy is recommended to investigate other causes such as cytomegalovirus. Simultaneous escalation of immunosuppression and/or FMT should be taken into account despite a lack and discordance of data, completing the antibiotic course over 10 days anyway [129,130]. In these flares, it is also important to evaluate cases with a multidisciplinary team (MDT), as has been demonstrated in a 2021 Chinese study, that ensures better outcomes [131].

## 7. Conclusions

Although more evidence is needed, available data suggest that IBD patients with an ongoing clinical flare should always be tested early for a possible *C. difficile* infection. In cases of confirmed positivity, patients could greatly benefit from the use of vancomycin or fidaxomicin as first-line options, also considering the addition of bezlotoxumab. In case of multiple recurrences, FMT via colonoscopy should be the treatment of choice since it allows to benefit from the effect on the dysbiotic profile of the gut microbiota and enhances the clinical picture in terms of IBD post-FMT [96,132].

In case patients are refractory to treatment, a multidisciplinary team (MDT) may be useful to tailor a different approach and to escalate, where appropriate, the anti-inflammatory/immunosuppressant therapies. Several lines of therapies for CDI are currently under investigation, including new antibiotics with a narrower spectrum of action, vaccines [111], and synthetic microbiome consortia [133]. As patients with IBD, especially those who are under immunosuppressive therapies, are fragile and have a non-negligible higher risk of infection and associated complications, future therapeutic strategies should be also targeting this population. In this context, synthetic microbial consortia are interesting not only as therapeutic tools for CDI but also from a research-based point of view since the experience in this setting may be translated into patients with IBD without concomitant CDI by identifying specific microbiome signatures that can contribute to inducing or maintaining remission if added to standard therapy, paving the way for microbiome-based precision medicine in IBD.

## Figures and Tables

**Figure 1 microorganisms-10-01315-f001:**
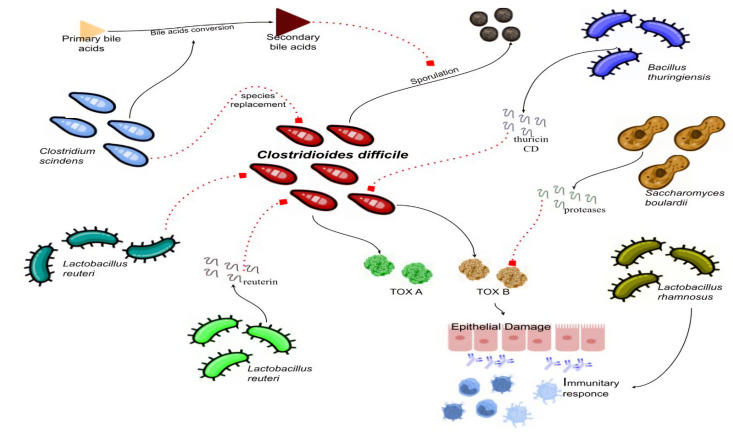
Possible effects of different bacterial strains on CDI infection. Continuous lines indicate stimulation; dashed lines indicate inhibition. CD—Clostridioides Difficile.

**Table 1 microorganisms-10-01315-t001:** Differences in risk factors, age of first presentation, and recurrence rates for CDI between IBD and general population. PPIs—proton-pump inhibitors, OR—odds ratio, HR—hazard ratio.

	IBD Population	General Population
**Impact of PPIs use on risk of CDI**	0.98 OR [0.54, 1.78] [15]	1.99 OR [1.73, 2.30] [16]
**Impact of Antibiotics use on risk of CDI**	1.85 OR [1.36, 2.52] [15]	3.55 OR (2.56–4.94) [17]
**Impact of Steroids use on risk of CDI**	0.96 [0.55, 1.69] [15]	1.81 OR (1.15–2.84) [18]
**Difference in hospital mortality in patients with CDI**	3.64 OR [2.66, 4.98] [15]	1.899 OR [1.269–2.840] [19]
**Difference in average age of first CDI presentation**	47.5 year for UC, 41 year for CD [14]	55 year [13]
**Difference in recurrence rates of CDI**	13% [12]	7% [12]
**HR for rCDI between IBD and not IBD population**	HR: 2.28; 95% CI: 1.16–4.48, *p* < 0.001 [12]	

**Table 2 microorganisms-10-01315-t002:** Keypoints on management of Clostridioides difficile infection (CDI) in adults with IBD. CDI—Clostridioides Difficile infection; TOX A/B—ELISA for detecting *C. difficile* toxin in fecal specimens, FMT—fecal microbiota transplantation.

In IBD population with diarrhea, always perform:	Clinical examination (asking them about recent antibiotics and corticosteroids use and registering vital signs) Stool cultures for enteroinvasive bacterial infections and *C. difficile* detection in feces (GDH and TOX A/B) Blood exams with hemoglobin and C-reactive protein [31]
If acute severe ulcerative colitis (ASUC) is suspected (>6 bloody stools per day and at least one among these systemic toxicity signs (temperature > 37.8 °C, pulse > 90 bpm, hemoglobin <105 g/L, or C-reactive protein > 30 mg/L), always perform:	Radiological imaging (CT) Sigmoidoscopy to re-staging and exclude CMV superinfection [31]
While stool cultures and exams for *C. difficile* detection are pending [31]	Do not delay corticosteroids treatment if ASUC is present
If *C. difficile* is confirmed [71]:	Suspend other antibiotics, if unnecessary Reduce dose of corticosteroids if they have been previously started In case of 1st or 2nd no fulminant episode, start Vancomycin or Fidaxomicin, adding Bezlotoxumab in high-risk subpopulations or in early recurrence (<6 months) In case of 2nd or subsequent recurrences, perform FMT if not contraindicated In case of Fulminant CDI (hypotension, shock, ileus, altered mental status, cardiorespiratory failure, lactic acidosis), always request surgical evaluation and radiological imaging. Always consider FMT via colonoscopy whenever the patient does not respond to standard antibiotic therapy or in severe presentation

**Table 3 microorganisms-10-01315-t003:** Dosages, mechanism of action, and clinical uses of the three antibiotics approved for CDI in IBD. IV—intravenous.

	Dosages	Mechanism of Action	Clinical Uses in CDI
**Fidaxomicin**	200 mg twice daily for 10 days per os, with or without	inhibition of RNA synthesis	No-fulminant episodes
**Vancomycin**	from 125 mg to 500 mg four times daily for 10–14 days per os	inhibition of bacterial cell wall synthesis	No-fulminant episodes Combined to metronidazole IV during fulminant episode without ileus
**Metronidazole**	500 mg three times daily for 10–14 days per os/IV	inhibition of nucleic acid synthesis	Per os if alternative agents are unavailable for unsevere episodes Intravenously Combined to Vancomycin during fulminant episode without ileus

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
