# Peer review of "Risk Factors, Diagnosis, and Management of Clostridioides difficile Infection in Patients with Inflammatory Bowel Disease"

_microorganisms, 2022, doi:10.3390/microorganisms10071315_

Round 1
Reviewer 1 Report
The manuscript reports on the comprehensive analysis of etiological, epidemiological, clinical, diagnostical and therapeutic aspects of Clostridioides difficile infection (CDI) as a complication of inflammatory bowel disease (IBD). This theme is of the great research and practice-oriented interest. The review describes known risk factors of CDI in patients with IBD, modern diagnostical assays and their reliability, treatment and prophylaxis of this severe complication. Special part of the review is devoted to managing IBD in patients with CDI. In whole, the review covers all the body of the latest papers and reports devoted to the CDI, especially in patients with IBD. The article is written in clear and professional manner. All misprints and minor inconsistencies in previous version have been corrected by the authors.
Author Response
The manuscript reports on the comprehensive analysis of etiological, epidemiological, clinical, diagnostical and therapeutic aspects of Clostridioides difficile infection (CDI) as a complication of inflammatory bowel disease (IBD). This theme is of the great research and practice-oriented interest. The review describes known risk factors of CDI in patients with IBD, modern diagnostical assays and their reliability, treatment and prophylaxis of this severe complication. Special part of the review is devoted to managing IBD in patients with CDI. In whole, the review covers all the body of the latest papers and reports devoted to the CDI, especially in patients with IBD. The article is written in clear and professional manner. All misprints and minor inconsistencies in previous version have been corrected by the authors.
R: We thank the reviewer for her/his time and effort in reviewing our paper, and are happy that our paper has caught her/his interest, and that he/she has appreciated the corrections we made
Reviewer 2 Report
An important area, with most points being potentially helpful to note. However, several edits and clarifications are needed.
1. Details of cited data on ‘increased risk’ for CDI with need further clarification; is greater risk of infection or of severity and outcomes?
2. Table 1 needs clearer explanation (especially of arrows) in text.
3. What other “opportunists” do IBD patients have (re line 82)?
4. Data should be cited re “any colonic involvement” …increased risk of CDI (line 110);
5. The paragraph in lines 157-160 is confusing and its message very unclear.
6. “quite atypical anamnesis” is unclear (line 168);
7. Conclusions need more careful editing: ex: line 435, should read something like “patients with IBD who have clinical flares”…SHOULD have CDI considered and sought as an important reason for their symptomatic ‘flare,’ as the treatment is quite different.” Removing “must” in most places should be changed to something less imperative, as every patient is different and the ‘proof’ is far from that absolute!
8. Similar dogmatic statements such as ‘required’ in line 187 should be modified to say should be considered. Overall, many points and evidence cited often have some validity, but are often overstated, and should be stated with greater caution throughout.
9. Many edits are needed, including; ‘exposition’ on line 41 in introduction should be exposure;
‘snd’ in line 63 in Table 1 legend;
On line 93, ‘cause’ should probably be ‘associated with’…
There are many others needing careful editing throughout.
Round 2
Reviewer 2 Report
Continued careful editing of the intended clarity in English is needed.
Though I cannot rewrite or edit this, examples that remain include several highlighted areas in lines:
79-81; with something more like "...capable of facilitating colonization and potentially infection with C. difficile...and alterations in gut microbiota." ie delete 'which are the main....
83; delete s
84 these patients
148-150; needs a careful rewrite. something lioke 'Although CDI with IBD does not appear to pose significant added risk of short term colectormy, ...from that of CDI without IBD, ... ie NOT from general population, right?!?
164; would change 'tested' to 'considered.'

Author Response
Please see the attachment

This manuscript is a resubmission of an earlier submission. The following is a list of the peer review reports and author responses from that submission.
Round 1
Reviewer 1 Report
The manuscript reports on the comprehensive analysis of etiological, epidemiological, clinical, diagnostical and therapeutic aspects of Clostridioides difficile infection (CDI) as a complication of inflammatory bowel disease (IBD). This theme is of the great research and practice-oriented interest. The review describes known risk factors of CDI in patients with IBD, modern diagnostical assays and their reliability, treatment and prophylaxis of this severe complication. Special part of the review is devoted to managing IBD in patients with CDI. In whole, the review covers all the body of the latest papers and reports devoted to the CDI, especially in patients with IBD. The article is written in clear and professional manner, but some misprints and minor inconsistencies need proper corrections.
Here I suggest some minor corrections concerning to misprints and some unclarities.
- Title. Line 2. Full name of the microorganism, Clostridioides difficile, in the title should be given. Moreover, here and there all Latin names of microbial species and genera should be printed using the italic style.
- Line 59. This is not a figure, but a table. In addition, the arrows in the table are not clear. It is very difficult to understand the sense of these arrows. For instance, the 3rd line of the table can mean that antibiotics exposure is shorter in patients with IBD than in general population, or antibiotics decrease risk of CDI in patients with IBD and increase it in general population. But in such case, should we consider recurrence rate, colectomy rate and onset presentation as risk factors of CDI also? No, as I see. In this case, consequently, the authors should give more precise legend to the table and explain here the sense of the arrows.
- Line 59. Do the authors mean “risk factors of IBD”? In such case, correct it.
- Line 63. “Dysregulation” is not appropriate term in this case, in my opinion. Because it means indirect impact to microbiota through a disorder of regulating functions provided by any system, nervous, endocrine, and other. But antibiotics do not produce indirect impact to microbiota, they directly kill or inhibit growth of the normal gut microbiota.
- Line 75. “Pathobiont” is again an improper term, because it means a pathogenic organism revealed in cases of disease only but which is absent in healthy persons. The best term for C. difficile describing it ability to persist asymptomatically is “opportunist”.
- Line 80. Misprint. Double “in”.
- Line 80. Misprint. Double “of”.
- Line 88. “In a recent metanalysis [26], the use of antibiotics doubled the risk of CDI in the IBD population.” This statement contradicts to Table 1. Correct it, or correct the table, or explain this inconsistency.
- Line 112. “Dysbiotic environment” is not clear and commonly used term. I recommend “dysbiotic state of gut” or simply “dysbiosis”.
- Line 118. “Bacteria” should be exchanged by “bacterium”.
- Line 130. “Proliferation” is an improper term, because it means reproduction. But it is a strong rule that one spore produces only one vegetative bacterial cell. So, I recommend using here terms “germination of spores” or “transformation of spores into vegetative cells”.
- Line 134. Misprint. Remove “in”.
- Line 147. “In the previous metaanalysis shows” should be exchanged by “The previous metaanalysis showed”.
- Line 285. Please, correct unclarity and duplication in the phrase “…because of its higher risk of considering their higher risk of severe CDI…”
- Line 304. Please, correct unclarity and duplication in the phrase “…because according there are data on some cardiologic exacerbations after infusion…”.
- Line 315. “Its” should be exchanged by “their”.
- Line 326. “Against” should be exchanged by “in cells of”, because it is a correct description of the mechanism of bacterial killing produced by the bacteriocin.
- Line 327. Add commas after “reuterin” and “peptide”.
- Line 343. Misprint. “c difficile” should be exchanged by “C. difficile”.
- Line 345. “Another field of interest is becoming the study and manipulation of mucosal biofilm.” Strange composition of the sentence that makes it unclear. I recommend to exchange it by “The study and manipulation of mucosal biofilm is becoming another field of interest.”
- Line 346. “An” should be exchanged by “a”.
- Line 347 and 352. Please, remove a repeat of the biofilm definition. “…surface associated microbial communities…”.
- Line 367. “Of sensitivity” should be exchanged by “in sensitivity”.
- Line 379. “Addiction” should be exchanged by “addition”.
Reviewer 2 Report
The authors comprehensively reviewed the diagnosis and management of C. difficile infection in IBD patients. It provided the updated and useful information to clinical physicians.
Reviewer 3 Report
Main comments:
- The authors aim to summarize the risk factors, diagnosis, and management of Clostridioides difficile infection in IBD patients. However, the manuscript mainly focuses on the first two parts, only a small section described the managing of CDI in patients with IBD at the very end of the text. Maybe the authors consider changing the title to more present the contents of the manuscript.
- Figure 1: It more looks like a table, not figure. Besides, what is the up and down arrow mean here? Since the general population was included here, what they compare with If it is mean increase and decrease?
Also, the IBD with CDI should include in this list. Please also consider the figure legend to clarify the contents.
- There are so many paragraphs (most of them are very short with just one sentence) which makes the whole manuscript more likely a notebook. It need re-organize in a logical and systematic way to make it easier to follow and read.
- Line 61: This section mainly describes the risk factors of IBD patients developing CDI, please revise the title to dispel the potential misleading.
- Figure 2: It more looks like a diagnostic trial.
- A working model chart to describe the correlation between IBD and CDI, and contents of this manuscript will be helpful.
- The authors may also consider add some future direction at the end of the paper.
Minor comments:
- Line 14-16: “…have a higher risk or CDI incidence…”, is it “of” not “or”? Please re-write the whole sentence which really hard to follow.
- Line 122: Please correct the reference citation format, and check the whole manuscript.